# Localizing neurosurgical instruments across domains and in the wild

**Markus Philipp**[1,2]                                             MARKUS.PHILIPP@ZEISS.COM

**Anna Alperovich**[3]                                               ANNA.ALPEROVICH@ZEISS.COM

**Marielena Gutt-Will**[4]                                          MARIELENA.GUTT-WILL@INSEL.CH

**Andrea Mathis**[4]                                                 ANDREA.MATHIS@INSEL.CH

**Stefan Saur**[2]                                                      STEFAN.SAUR@ZEISS.COM

**Andreas Raabe**[4]                                                 ANDREAS.RAABE@INSEL.CH

**Franziska Mathis-Ullrich**[1]                                    FRANZISKA.ULLRICH@KIT.EDU

[1] *Institute for Anthropomatics and Robotics, Karlsruhe Institute of Technology, Germany*

[2] *Carl Zeiss Meditec AG, Oberkochen, Germany*

[3] *Carl Zeiss AG, Oberkochen, Germany*

[4] *University Hospital of Bern, Switzerland*

**Editors:** Under Review for MIDL 2021

## Abstract

Towards computer-assisted neurosurgery, robust methods for instrument localization on neurosurgical microscope video data are needed. Specifically for neurosurgical data, challenges arise from visual conditions such as strong blur and from an unknowingly large variety of instrument types. For neurosurgical domain, instrument localization methods must generalize across different sub-disciplines such as cranial tumor and aneurysm surgeries which exhibit different visual properties. We present and evaluate a methodology towards robust instrument tip localization for neurosurgical microscope data, formulated as coarse saliency prediction. For our analysis, we build a comprehensive dataset comprising *in-the-wild* data from several neurosurgical sub-disciplines as well as phantom surgeries. Comparing single stream networks using either image or optical flow information, we find complementary performance of both networks. Plain optical flow enables better cross-domain generalization, while the image-based network performs better on surgeries from the training domain. Based on these findings, we present a two-stream architecture that fuses image and optical flow information to utilize the complementary performance of both. Being trained on tumor surgeries, our architecture outperforms both single stream networks and shows improved robustness on data from different neurosurgical sub-disciplines. From our findings, future work must focus more on how to incorporate optical flow information into fusion architectures to further improve cross-domain generalization.

**Keywords:** Instrument localization, neurosurgery, microscope, robust, cross-domain, saliency

## 1. Introduction

Each year, more than 13.8 million neurosurgical interventions are needed worldwide (Dewan et al., 2018). Neurosurgeons require surgical microscopes for treating fine anatomical structures in the brain or spine. Algorithms for automatic identification of the surgeon's regions of interest from microscope videos become key ingredient towards computer-assisted

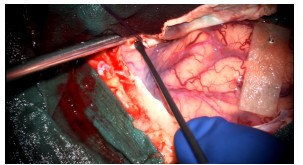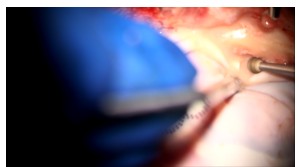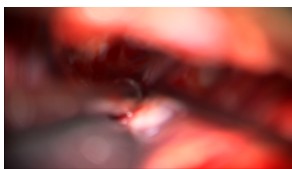

Figure 1: Situations in our neurosurgical dataset, illustrating the difficulty of instrument localization due to variety of instruments and visual properties (blur, reflections).

neurosurgery. Tips of surgical instruments were identified as a major region of interest in the microscope view through eye tracking studies with neurosurgeons (Eivazi et al., 2012). Developing algorithms for robust neurosurgical instrument localization needs to tackle both, the unknowingly large variety of instrument shapes as well as challenging visual conditions due to reflections and blur (Fig. 1). As visual conditions between e.g. cranial tumors and aneurysms can vary significantly, generalization across these data domains is crucial.

In contrast to laparoscopic data (Ross et al., 2020) only few approaches for instrument localization exist for neurosurgery (Bouget et al., 2015; Kalavakonda et al., 2019). Following our goal to detect a surgeon's regions of interest we focus on instrument tips and abstain from semantic segmentation of complete instruments in order to achieve real-time capability. However, annotations of instrument tips are inherently *fuzzier* than pixel-wise segmentation masks, as the tip definition depends on the individual instrument shape. We propose to incorporate this annotation fuzziness by defining a *soft* localization problem instead of bounding box prediction (Rieke et al., 2016). Inspired by Islam et al. (2019), who included saliency prediction into a multi-task problem to support semantic instrument segmentation, we propose saliency learning as primary task. Following non-medical saliency literature for dynamic scenes (Bak et al., 2018), we incorporate optical flow to capture instrument-agnostic, characteristic temporal variations caused by instruments. We consider a coarse saliency learning problem, assuming that regions of interest per definition are not on a pixel resolution. Additionally, coarse saliency maps do not suffer from flickering artifacts when applied to videos and can be used in real-time. We solve saliency prediction as regression problem where the prediction corresponds to uncertainty of instrument presence.

**Contributions.** We present a methodology towards robust instrument tip localization in neurosurgical microscope video data. First, we analyze robustness and generalization capabilities of single stream convolutional neural networks (CNN) using either image or optical flow information as an input. Second, based on our findings, we propose a spatio-temporal two-stream CNN approach. Ensuring a well-validated methodology, we build our analyses on a clinical dataset containing *complete*, *randomly chosen* (i.e. in the wild) cranial tumor, vascular, and spine surgeries. Furthermore, we include phantom data, representing a larger domain shift, and thus imposing a challenge for cross-domain generalization.

## 2. Methodology

We formulate instrument tip localization as predicting a coarse saliency map $Q_{pred} = (p_{i,j}) \in \mathbb{R}^{n \times m}$, with probability $p_{i,j} \in [0, 1]$ for a pixel $(i, j)$ to show an instrument tip. In our work, we compute saliency maps with $n = 9$, $m = 16$ (Fig. 2). By learning probabilities $p_{i,j}$ we

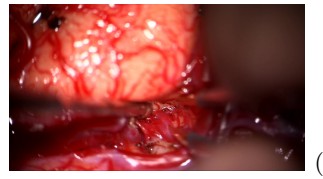 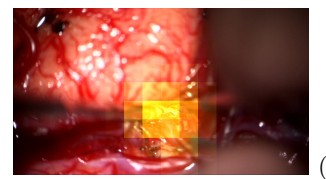 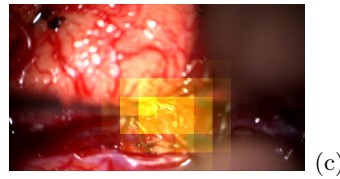

Figure 2: (a) Scene, two instruments. (b) Ground truth. (c) Predicted saliency, SIM=0.8.

incorporate the instrument tip ambiguity as the tip definition depends on the instrument shape. As evaluation metric we use similarity or histogram intersection (SIM) of $Q_{pred}$ to ground truth $Q_{GT}$: $\text{SIM} = \sum_{i,j} \min(Q_{GT}, Q_{pred}) \in [0, 1]$ where $\sum_{i,j} Q_{GT} = \sum_{i,j} Q_{pred} = 1$.

Building on a comprehensive dataset, we first analyze single stream network performance, i.e., a CNN for saliency prediction using *either* image information *or* optical flow as input. Optical flow input represents a 2D-vector field, describing the apparent motion between the current and the previous video frame. In our experiments, we estimate optical flow by using PWC-Net (Sun et al., 2018) (see Appendix A).

### 2.1. Dataset collection

Video recordings from 10 cranial tumor, 2 cranial vascular and 2 spine surgeries with approx. 20 different instruments were collected at the University Hospital of Bern with a surgical microscope (ZEISS KINEVO 900). Using images from the *complete surgery duration*, we refer to our data as clinical *in-the-wild* data. Qualitatively, we observed domain gaps (e.g. level of blur, instrument types) between the tumor, vascular and spinal surgeries. Enforcing significantly larger domain gaps to this clinical data, we recorded videos using an UpSim phantom (*UpSim Neurosurgical Box*) under the same microscope in our lab (Appendix B).

Annotation of video data (1 Hz) was done by four non-medical annotators in a procedure developed with expert neurosurgeons. Every image was seen by three of the annotators. Annotator 1 (A1) labels whether instruments tips are fully visible and, if so, draws a bounding box centered and encompassing each entire tip. A2 verifies and corrects these bounding boxes. Independently from A1, A3 labels whether instrument tips are fully visible, allowing consensus check with A1. Frames without or only partly visible tips were excluded.

For converting bounding box annotations to saliency maps, we perform label smoothing using Gaussian sampling. As the definition of the tip is instrument shape specific, this label smoothing is beneficial to compensate for natural tip location ambiguity. We define a training dataset with tumor surgeries (*TUMOR*) and another with phantom surgeries (*PHANTOM*); testing is done on unseen cases from all available domains (Tab. 1).

Table 1: (a) Single-domain datasets *TUMOR*, *PHANTOM* for training. (b) Cross-domain generalization tested on cranial tumor, cranial vascular and spinal (i.e., clinical data) and phantom surgeries. Legend: (# surgeries/total # annotated images).

| | Name setting | Training data | Validation data |
|---|---|---|---|
| (a) | TUMOR | tumor (6/22315) | tumor (2/5093) |
| | PHANTOM | phantom (4/884) | phantom (2/475) |

| | Test data |
|---|---|
| (b) | tumor (2/6489), vascular (2/13601), spine (2/8305), phantom (2/482) |

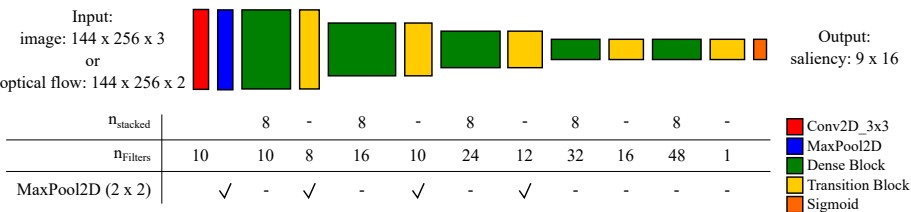

Figure 3: Single stream CNN inspired by DenseNet (Huang et al., 2017), with five building blocks (see parameterization). Every Dense block follows DenseNet-BC design.

## 2.2. Single stream network analysis

We compare two single stream saliency prediction networks with same architecture (Fig. 3). The first network (IMG) takes image information as input, while the second (OF) uses optical flow. We train both networks on *TUMOR* and *PHANTOM* data separately. The influence of network input and training domain is analyzed w.r.t. cross-domain generalization and robustness. Generalization is investigated by surgery-wise SIM distribution (Fig. 4). Robustness represents deviation from mean value for every single sample (Fig. 5). For clarity, here we plot only one case for each domain.

From our analysis we conclude: (1) IMG performs better than OF for identical training and test domains. (2) IMG has larger relative performance variation across different domains than OF. (3) IMG and OF are often complementary, especially if one of both performs poorly.

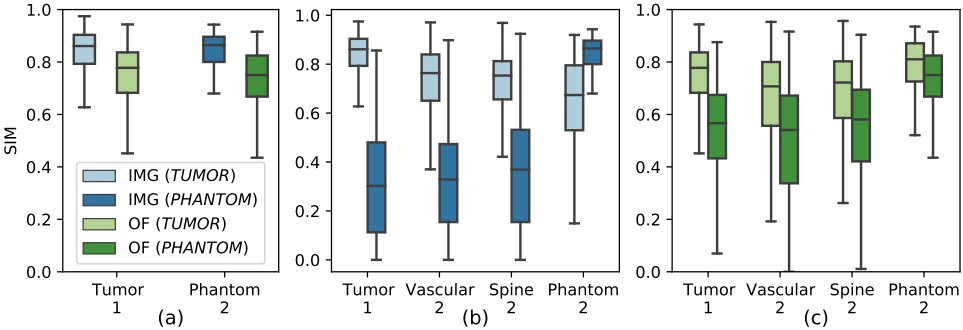

Figure 4: SIM distribution for test surgeries (tumor case 1, vascular case 2, ...) without outliers for IMG, OF trained on *TUMOR* or *PHANTOM*. (a) IMG exhibits higher in-domain performance than OF. (b) IMG trained on *TUMOR* (=IMG(*TUMOR*)) shows performance drop when tested on other domains. IMG(*PHANTOM*) displays poor generalization on clinical data (tumor, vascular, spine). (c) OF shows better cross-domain generalization than IMG, compare (b).

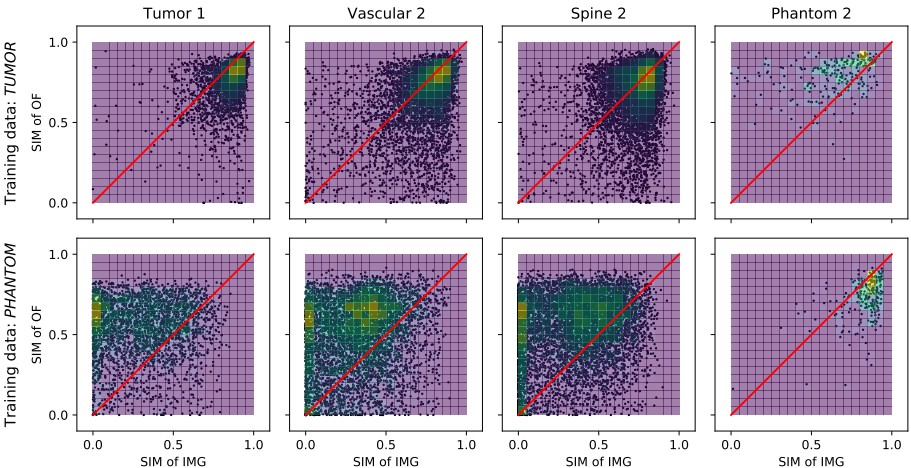

Figure 5: To investigate robustness, we analyze the distribution of all test images based on a scatter plot with (x,y) = (SIM$_{\text{IMG}}$, SIM$_{\text{OF}}$) and density overlay. The red reference line indicates identical performance of both networks. Ideally, there is no deviation (i.e. robustness) and all scatters are located in the up-right corner. When training on *TUMOR*, we find a broad distribution on both sides of the reference line, indicating complementary performance for `IMG` and `OF`. When trained on *PHANTOM*, `OF` outperforms `IMG`. However, we still observe complementary performance, where scatters are distributed on both sides of the reference line.

## 2.3. Spatio-temporal fusion two-stream network approach

Based on the analysis, we propose a two-stream fusion architecture `FUS`, where both image and optical flow are model inputs. Leveraging the complementary performance of the two single stream networks, we enable our architecture to exploit all available information from both inputs. To extract deep features from both input modalities, two encoder pathways are combined only when reaching final feature resolution (Fig. 6).

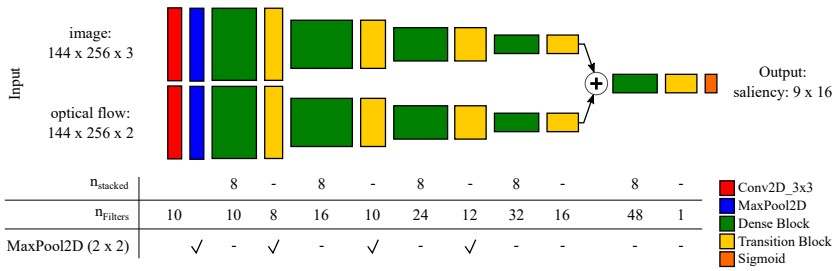

Figure 6: Two-stream fusion network `FUS` with encoders having the same building blocks as in the single stream networks. Parameters are the same for both pathways. Fusion is done by adding feature maps, avoiding increase of model complexity.

### 2.4. Training and implementation

All experiments are conducted with same settings. Inputs are sized $256 \times 144$. Optical flow is pre-computed in Cartesian representation. Data augmentation consists of spatial and temporal random crop, flip, rotation offset (only optical flow), random contrast, color and brightness (only image). Both inputs are normalized w.r.t. mean and standard deviation. Loss is mean-squared error. Training is performed from scratch with Adam optimizer and an initial learning rate of 0.01. Learning rate was decayed with rate 0.1 based on plateau detection of the validation SIM (on same domain data) with patience $= 50$ until $10^{-6}$. Early stopping was included on the validation SIM with patience $= 100$. Models are trained on Intel i9-9900 with 64 GB RAM and NVIDIA RTX 2080 SUPER. The longest training took 12h. Inference time for given image and optical flow is $<50$ ms.

### 3. Evaluation

We analyze cross-domain generalization using SIM mean comparison (Tab. 2). Our `FUS` architecture achieves the best performance on all clinical data when trained on *TUMOR* (sample predictions for `FUS` see Appendix C). When tested on phantom data, `FUS` is better than `IMG` but falls behind `OF`. This confirms that optical flow information supports our network to generalize well on large domain shifts. Although both `IMG` and `FUS` overfit when trained on *PHANTOM*, `FUS` seems to benefit from optical flow when testing on clinical data. To avoid focusing on mean values only, we perform quantile distribution analysis to verify robustness (Fig. 7). When trained on *TUMOR*, increased robustness on clinical test cases for `FUS` over `IMG` and `OF` can be observed. When training on *PHANTOM*, large domain shifts impose challenges for all networks w.r.t. robustness. Presumably, training `FUS` on *PHANTOM* focuses too much on image information. When tested on a domain different from training, optical flow improves robustness of the `FUS` architecture over the relatively poor `IMG` performance. Similar to robustness verification in Fig. 5, we conduct a single-

Table 2: Mean values of SIM and pairwise t-tests ($\alpha < 0.05$) with Bonferroni correction. Best algorithm in **bold**. Legend: ** : $<0.001$, * : $<0.05$ (after correction).

| | | Tumor 1 | Tumor 2 | Vascular 1 | Vascular 2 | Spine 1 | Spine 2 | Phantom 1 | Phantom 2 |
|---|---|---|---|---|---|---|---|---|---|
| | IMG | 0.830 | 0.808 | 0.784 | 0.716 | 0.784 | 0.718 | 0.728 | 0.634 |
| | OF | 0.741 | 0.727 | 0.695 | 0.650 | 0.732 | 0.670 | **0.813** | **0.788** |
| Training on | FUS | **0.840** | **0.832** | **0.800** | **0.740** | **0.805** | **0.765** | 0.770 | 0.712 |
| *TUMOR* | $p_{IMG=OF}$ | ** | ** | ** | ** | ** | ** | ** | ** |
| | $p_{IMG=FUS}$ | ** | ** | ** | ** | ** | ** | ** | ** |
| | $p_{OF=FUS}$ | ** | ** | ** | ** | ** | ** | ** | ** |
| | IMG | 0.310 | 0.373 | 0.388 | 0.328 | 0.345 | 0.355 | 0.846 | 0.827 |
| | OF | **0.535** | **0.530** | **0.492** | **0.496** | **0.564** | **0.540** | 0.736 | 0.727 |
| Training on | FUS | 0.372 | 0.398 | 0.364 | 0.386 | 0.411 | 0.416 | 0.853 | **0.843** |
| *PHANTOM* | $p_{IMG=OF}$ | ** | ** | ** | ** | ** | ** | ** | ** |
| | $p_{IMG=FUS}$ | ** | ** | ** | ** | ** | ** | | ** |
| | $p_{OF=FUS}$ | ** | ** | ** | ** | ** | ** | ** | ** |

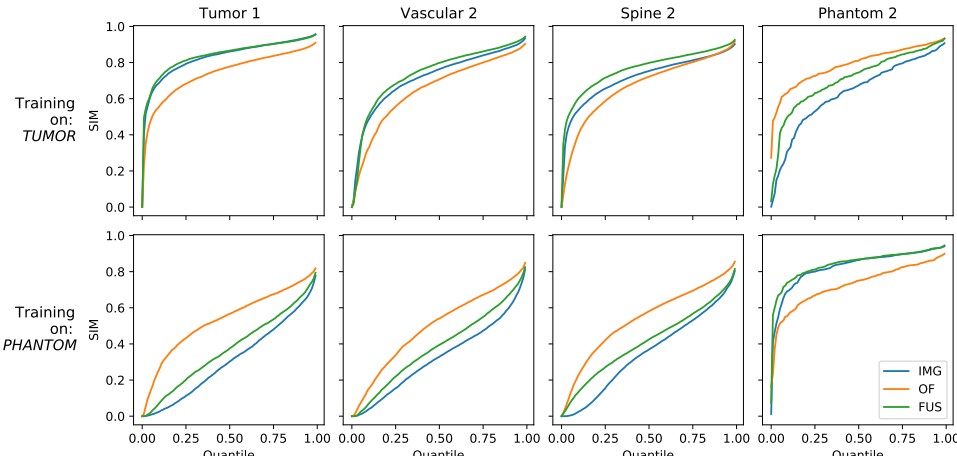

Figure 7: Quantile analysis showing robustness. The area under the curve being 1 indicates best performance for all samples from a domain. Upper row: When trained on *TUMOR*, `FUS` shows better robustness than `IMG` and `OF` on clinical data (tumor, vascular, spine). Lower row: Although none of the networks are robust on clinical data when trained on *PHANTOM*, optical flow information improves robustness of `FUS` over `IMG`.

sample analysis to investigate when `FUS` improves over `IMG` and `OF` (Fig. 8). When `FUS` is trained on *TUMOR*, it tendentially improves when one of the single stream networks performs poorly, indicating that `FUS` exploits complementary behavior of `IMG` and `OF`. Analysis if `FUS` improves over `IMG` and `OF` simultaneously, however, revealed no such synergy.

## 4. Discussion and Conclusions

In a real-world neurosurgery scenario for instrument localization, one does not know which data to expect next. Based on our analysis, we conclude that both modalities, image and optical flow, have to be present as network inputs. Thus, we developed a two-stream architecture to achieve a robust and generic solution. To ensure exploitation of relevant features from both input modalities, we fuse the encoder pathways only at a late stage. Trained on tumor surgeries, our architecture shows best results on other clinical data compared to the single stream networks. To simulate large domain shifts we train on phantom surgeries and evaluate on clinical data. We observe improved performance of our architecture compared to the purely image-based network. This observation supports the idea that optical flow contains essential information, when the image context is not familiar to the network. Since our solution has to work in the wild, it is necessary to have it reliable irrespective of imaging and clinical conditions. Our results show that when extracting enough information from both input modalities it is possible to fulfill these requirements. Future work will investigate improved fusing architectures and the role of image and optical flow information. We believe that solutions for instrument localization must stronger incorporate optical flow to ensure performance on unseen domains.

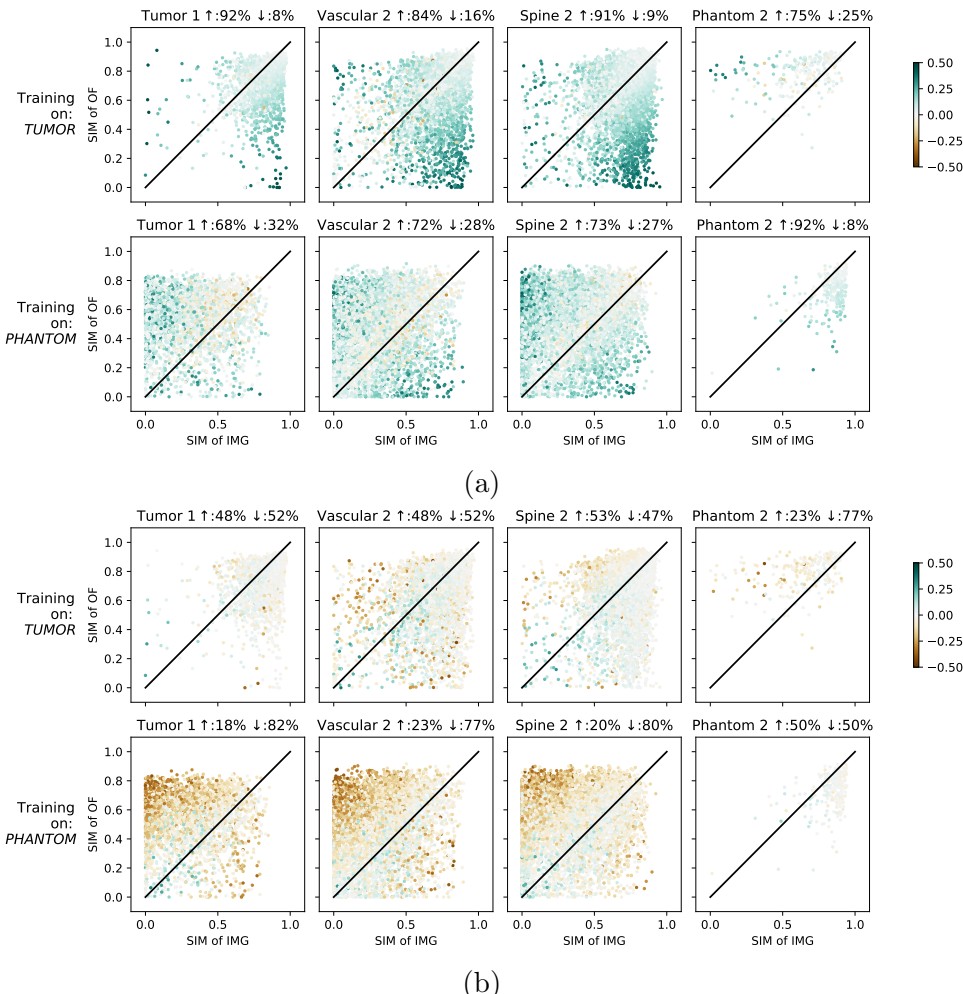

Figure 8: (a) To analyze improvement of FUS over IMG or OF, the scatters are colored with $c = \mathrm{SIM_{FUS}} - min(\mathrm{SIM_{IMG}}, \mathrm{SIM_{OF}})$. Green scatters indicate FUS improves at least over one of the networks (↑: % of scatters with $c > 0$). Yellow means deterioration compared to both. When training FUS on $TUMOR$, green samples on both sides of the reference line indicate FUS benefits from the complementary behavior. When training FUS on $PHANTOM$, FUS profits from both input modalities on clinical data in at least 68% of samples. However, we also find single yellow scatters for clinical data where FUS slightly deteriorates. (b) shows when FUS improves over IMG and OF simultaneously (coloring: $c = \mathrm{SIM_{FUS}} - max(\mathrm{SIM_{IMG}}, \mathrm{SIM_{OF}})$). Green indicates FUS improves over both networks simultaneously. Yellow means FUS performs worse than the best of both. For both training datasets, the few observed green points indicate that FUS could not leverage additional synergy effects (from image and optical flow data, respectively).

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

## Appendix A.  Optical flow

Optical flow was estimated using PWC-Net (Sun et al., 2018). We show sample plots of the estimated optical flow in Fig. 9.

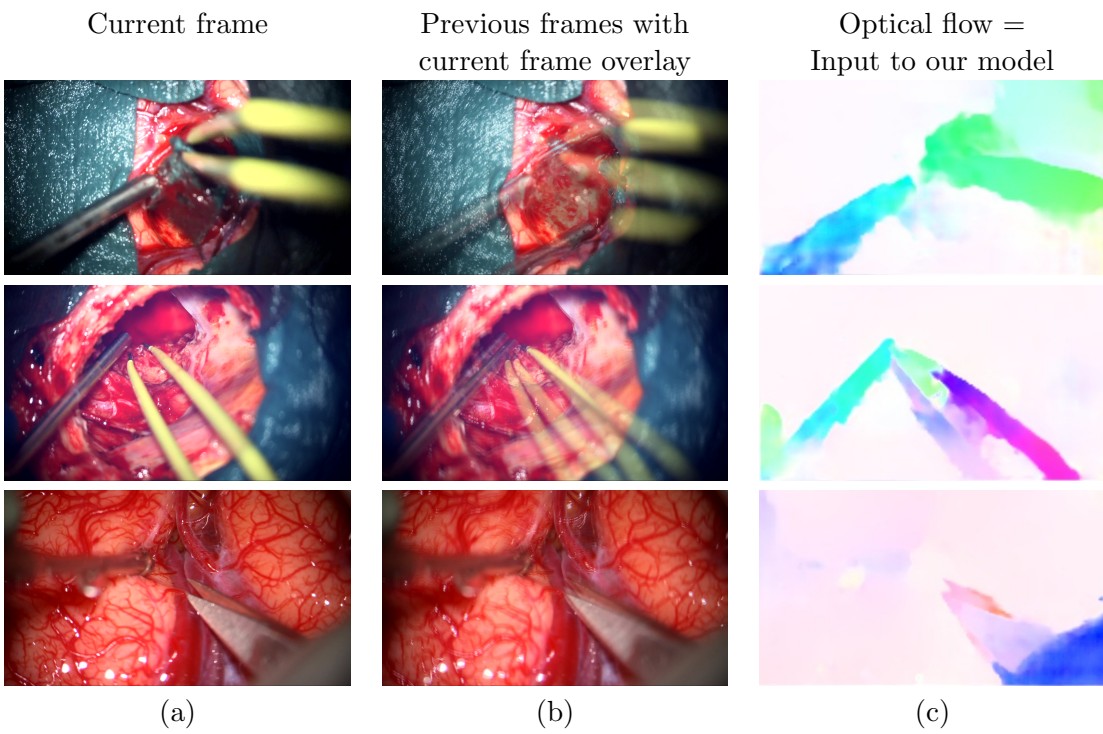

| Current frame | Previous frames with current frame overlay | Optical flow = Input to our model |
| :---: | :---: | :---: |
| (a) | (b) | (c) |

Figure 9: Sample images with estimated optical flow. Each row refers to a video sequence. Column (a) shows the current video frame $I_{t=0}$. Column (b) shows the previous frame $I_{t=-1}$ with transparent overlay of the current frame $I_{t=0}$ to highlight motions. Column (c) shows the corresponding optical flow between $I_{t=0}$ and $I_{t=-1}$. While optical flow is estimated as Cartesian vector field $(v_x, v_y)$, we convert $(v_x, v_y)$ to polar space (mag, ang) for displaying. We plot the optical flow using HSV color space, where the hue denotes angle and saturation shows the (normalized) vector magnitude.

## Appendix B. Phantom data

Phantom data was recorded using an UpSim Neurosurgical Box under a ZEISS KINEVO 900. The UpSim Neurosurgical Box was developed by neurosurgeons for training. In our study we use a variety of widely used neurosurgical instruments: two different suctions, two different forceps, monopolar, tweezers. Optical settings (zoom, focus) were varied within the sequences. Each frame shows two instruments. Sample images are shown in Fig. 10.

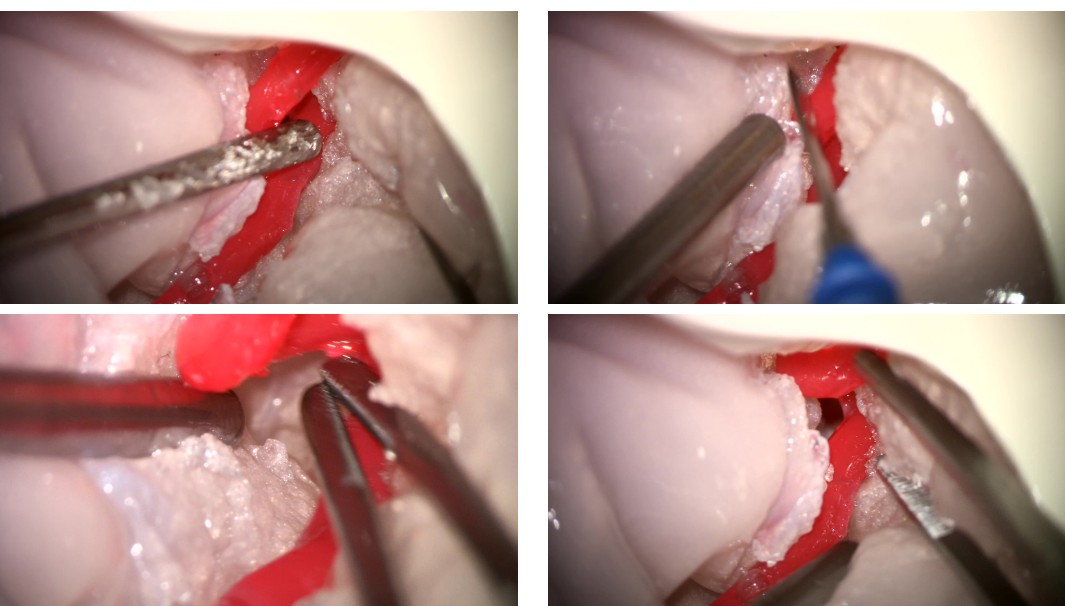

Figure 10: Sample images from phantom recordings.

## Appendix C. Sample images with ground truth and predictions

We show example predictions from the proposed architecture together with ground truth annotations from the clinical test surgeries. For each prediction we provide SIM and L2 metric. Samples are selected to display the range of good to poor predictions: good performance (Fig. 11), medium performance (Fig. 12), poor performance (Fig. 13).

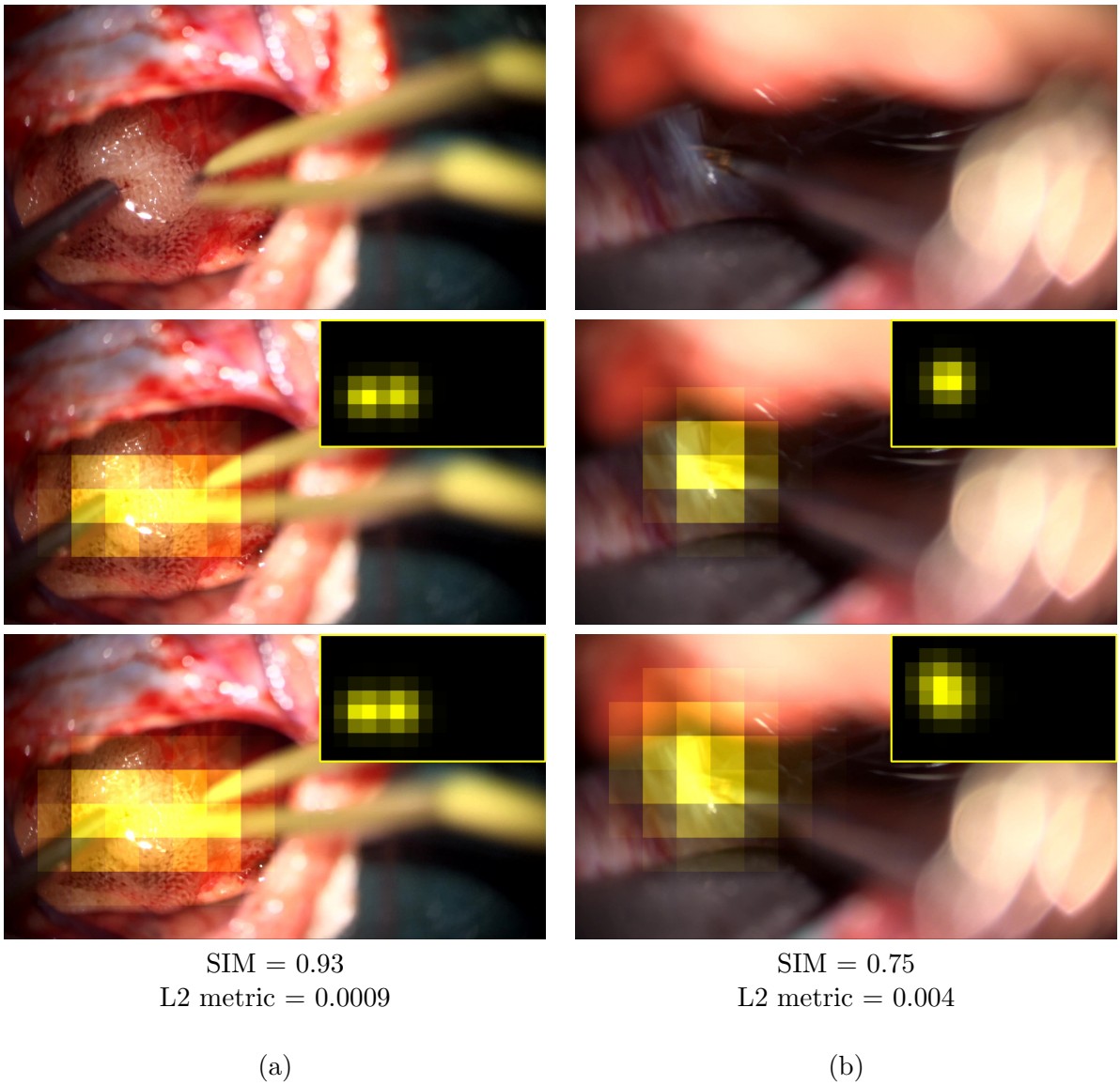

SIM = 0.93                    SIM = 0.75
L2 metric = 0.0009            L2 metric = 0.004

(a)                          (b)

Figure 11: Two video frames from different surgeries where our model shows good performance. Top row: image, middle row: saliency ground truth overlaid over image with plain saliency map in the top right corner, low: saliency prediction.

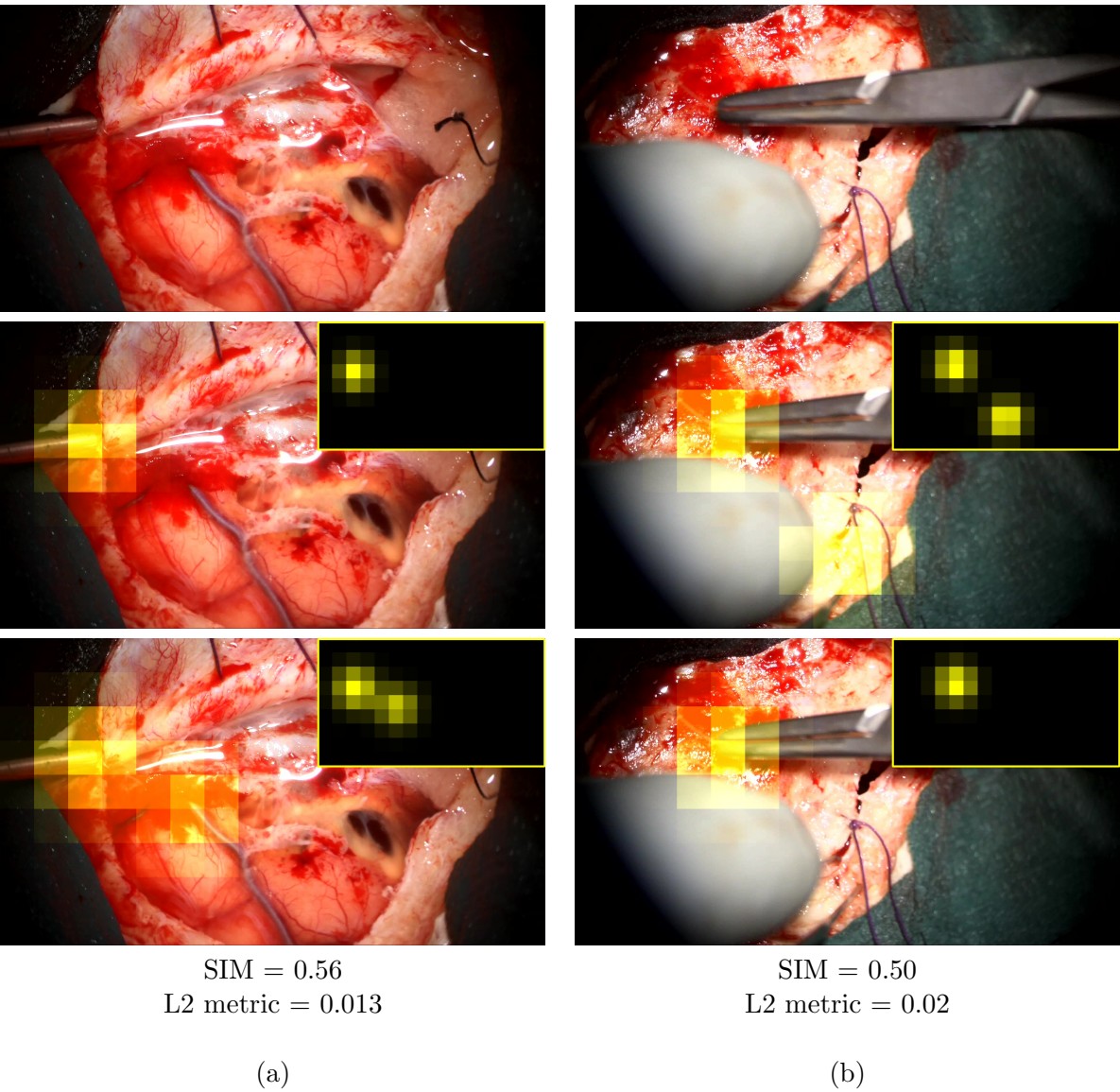



SIM = 0.56           SIM = 0.50
L2 metric = 0.013        L2 metric = 0.02

(a)              (b)



Figure 12: Two video frames from different surgeries where our model shows medium performance. Top row: image, middle row: saliency ground truth overlaid over image with plain saliency map in the top right corner, low: saliency prediction.

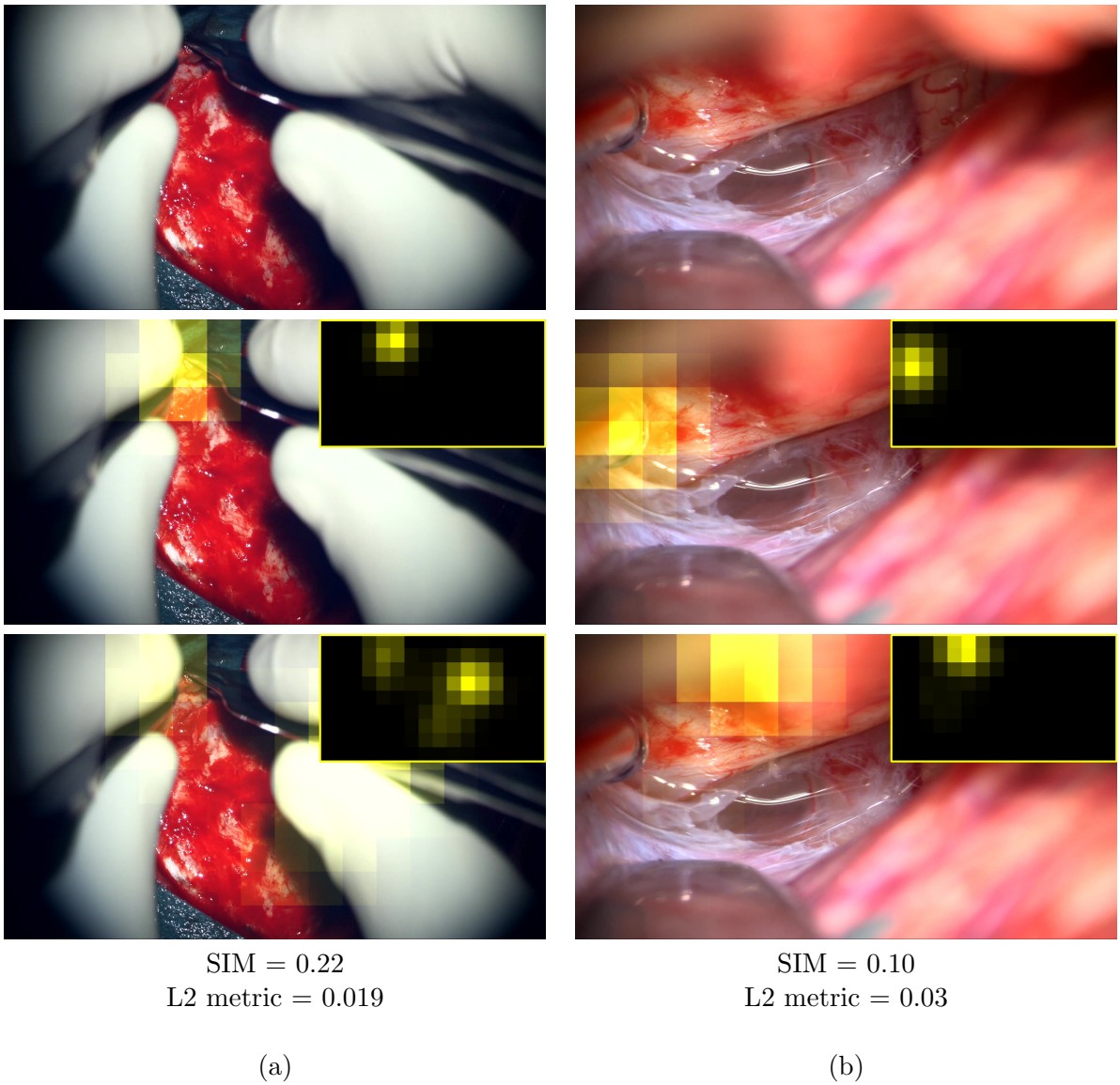

SIM = 0.22
L2 metric = 0.019

SIM = 0.10
L2 metric = 0.03

(a)                                                                 (b)

Figure 13: Two video frames from different surgeries where our model shows poor performance. Top row: image, middle row: saliency ground truth overlaid over image with plain saliency map in the top right corner, low: saliency prediction.

## Appendix D. Numerical evaluation

Additionally to Tab. 2, we provide the median values for the SIM distributions (Tab. 3).

Table 3: Legend: $\mu$ : Mean, M : Median, ** : $p < 0.001$, * : $p < 0.05$ (both corrected). For comparison, pairwise t-tests ($\alpha < 0.05$) with Bonferroni correction were used. Largest value in **bold**. Abbreviation: T - tumor, V - vascular, etc.

| | | T 1 | T 2 | V 1 | V 2 | S 1 | S 2 | P 1 | P 2 |
|---|---|---|---|---|---|---|---|---|---|
| colspan9: **Training data : _TUMOR_** |
| IMG | $\mu$ | 0.830 | 0.808 | 0.784 | 0.716 | 0.784 | 0.718 | 0.728 | 0.634 |
| | M | 0.861 | 0.851 | 0.808 | 0.763 | 0.824 | 0.753 | 0.753 | 0.674 |
| OF | $\mu$ | 0.741 | 0.727 | 0.695 | 0.650 | 0.732 | 0.670 | **0.813** | **0.788** |
| | M | 0.778 | 0.773 | 0.744 | 0.707 | 0.767 | 0.722 | **0.834** | **0.810** |
| FUS | $\mu$ | **0.840** | **0.832** | **0.800** | **0.740** | **0.805** | **0.765** | 0.770 | 0.712 |
| | M | **0.866** | **0.865** | **0.825** | **0.798** | **0.835** | **0.798** | 0.787 | 0.744 |
| $p_{IMG=OF}$ | | ** | ** | ** | ** | ** | ** | ** | ** |
| $p_{IMG=Fusion}$ | | ** | ** | ** | ** | ** | ** | ** | ** |
| $p_{OF=Fusion}$ | | ** | ** | ** | ** | ** | ** | ** | ** |
| colspan9: **Training data : _PHANTOM_** |
| | | T 1 | T 2 | V 1 | V 2 | S 1 | S 2 | P 1 | P 2 |
| IMG | $\mu$ | 0.310 | 0.373 | 0.388 | 0.328 | 0.345 | 0.355 | 0.846 | 0.827 |
| | M | 0.302 | 0.368 | 0.389 | 0.329 | 0.345 | 0.369 | 0.874 | 0.864 |
| OF | $\mu$ | **0.535** | **0.530** | **0.492** | **0.496** | **0.564** | **0.540** | 0.736 | 0.727 |
| | M | **0.566** | **0.566** | **0.529** | **0.541** | **0.613** | **0.581** | 0.748 | 0.750 |
| FUS | $\mu$ | 0.372 | 0.398 | 0.364 | 0.386 | 0.411 | 0.416 | 0.853 | **0.843** |
| | M | 0.376 | 0.401 | 0.355 | 0.395 | 0.416 | 0.425 | 0.868 | **0.868** |
| $p_{IMG=OF}$ | | ** | ** | ** | ** | ** | ** | ** | ** |
| $p_{IMG=Fusion}$ | | ** | ** | ** | ** | ** | ** | | ** |
| $p_{OF=Fusion}$ | | ** | ** | ** | ** | ** | ** | ** | ** |

