# OpenReview forum: "Localizing neurosurgical instruments across domains and in the wild"
_MIDL.io/2021/Conference — MIDL 2021_

### Meta-Review · Area_Chairs · 2021-03-29

**Recommendation:** Accept (Poster)

**Metareview:**

The authors present and thoroughly evaluate a method for localizing instruments for neurosurgery. The authors present the work as a validation/application paper, and combine image and optical flow in a network-based approach.

The reviewers had mixed feelings about the scope and novelty of the paper, mostly categorizing the paper as methodological or both (methods and application), while the authors meant it as an application (it seems). This led to several comments about the lack of novelty or some unclear results (such as the cross-domain). The reviewers also had clarifying questions, such as about the optical flow data. As it stood, the submission was very borderline.

The authors gave good responses and addressed most (although not all) of the main concerns of the reviewers. All reviewers stood by their weak accept ratings. Overall, I think this is a straightforward application paper with a worthwhile application that the reviewers thought warranted discussion at MIDL. I recommend a borderline accept, and emphasize that the authors should take all the thorough reviews into account for the camera ready, which will make sure the paper is a worthwhile addition to MIDL.

**Paper Type:**

validation/application paper

---

### Decision · Program_Chairs · 2021-03-31

Accept